# Extracellular Vesicles from SOD3-Transduced Stem Cells Exhibit Improved Immunomodulatory Abilities in the Murine Dermatitis Model

**DOI:** 10.3390/antiox9111165

**Published:** 2020-11-23

**Authors:** Ji Won Yang, Yoojin Seo, Tae-Hoon Shin, Ji-Su Ahn, Su-Jeong Oh, Ye Young Shin, Min-Jung Kang, Byung-Chul Lee, Seunghee Lee, Kyung-Sun Kang, Jin Hur, Yeon-Soo Kim, Tae-Yoon Kim, Hyung-Sik Kim

**Affiliations:** 1Department of Life Science in Dentistry, School of Dentistry, Pusan National University, Yangsan 50612, Korea; midnightnyou@naver.com (J.W.Y.); anjs08@naver.com (J.-S.A.); dhtnwjd26@naver.com (S.-J.O.); bubu3935@naver.com (Y.Y.S.); 2Dental and Life Science Institute, Pusan National University, Yangsan 50612, Korea; amaicat24@naver.com (Y.S.); kkang085@naver.com (M.-J.K.); 3Translational Stem Cell Biology Branch, National Heart, Lung and Blood Institute, National Institutes of Health, Bethesda, MD 20892, USA; thshin1125@gmail.com (T.-H.S.); tlc0502@snu.ac.kr (B.-C.L.); 4Institute for Stem Cell and Regenerative Medicine in Kangstem Biotech, Biomedical Science Building, Seoul National University, Seoul 08826, Korea; leesh@kangstem.com (S.L.); kangpub@snu.ac.kr (K.-S.K.); 5Adult Stem Cell Research Center and Research, Institute for Veterinary Science, College of Veterinary Medicine, Seoul National University, Seoul 08826, Korea; 6Department of Convergence Medicine, Pusan National University School of Medicine, Yangsan 50612, Korea; gene44@gmail.com; 7Graduate School of New Drug Discovery & Development, Chungnam National University, Daejeon 34134, Korea; kimys58@cnu.ac.kr; 8Department of Dermatology, Seoul St. Mary’s Hospital, College of Medicine, The Catholic University of Korea, Seoul 06591, Korea

**Keywords:** stem cell, antioxidant enzyme, superoxide dismutase 3, immunomodulation, dermatitis

## Abstract

The immunoregulatory abilities of mesenchymal stem cells (MSCs) have been investigated in various autoimmune and allergic diseases. However, the therapeutic benefits observed in preclinical settings have not been reproducible in clinical trials. This discrepancy is due to insufficient efficacy of MSCs in harsh microenvironments, as well as batch-dependent variability in potency. Therefore, to achieve more beneficial and uniform outcomes, novel strategies are required to potentiate the therapeutic effect of MSCs. One of simple strategies to augment cellular function is genetic manipulation. Several studies showed that transduction of antioxidant enzyme into cells can increase anti-inflammatory effects. Therefore, we evaluated the immunoregulatory abilities of MSCs introduced with extracellular superoxide dismutase 3 (SOD3) in the present study. SOD3-overexpressed MSCs (SOD3-MSCs) reduced the symptoms of murine model of atopic dermatitis (AD)-like inflammation, as well as the differentiation and activation of various immune cells involved in AD progression. Interestingly, extracellular vesicles (EVs) isolated from SOD3-MSCs delivered SOD3 protein. EVs carrying SOD3 also exerted improved therapeutic efficacy, as observed in their parent cells. These results suggest that MSCs transduced with SOD3, an antioxidant enzyme, as well as EVs isolated from modified cells, might be developed as a promising cell-based therapeutics for inflammatory disorders.

## 1. Introduction

Mesenchymal stem cells (MSCs) are capable of self-renewal and multipotent differentiation, which make them a promising regenerative therapeutics in tissue engineering [1]. Moreover, MSCs also exhibit low immunogenicity and immunomodulatory properties [2,3]. Recent reports, including our own studies, have shown that MSCs from various sources can exert protective or therapeutic efficacy against a variety of immune disorders [4,5,6,7,8,9,10,11,12]. Despite these benefits in preclinical reports, the efficacy of MSCs in clinical trials remains controversial because of their unsatisfactory outcomes. To overcome this limitation, researchers tried to augment the therapeutic function of MSCs through various strategies, such as cell priming, genetic modification, spheroid formation, and combination with biomaterials or synergistic drugs [11,13,14,15,16]. Among these strategies, genetic manipulation of MSCs has advantages in that the mechanistic target is clear and specific.

Oxidative stress results from an imbalance between the production of reactive oxygen species (ROS) and their elimination by protective mechanisms, which leads to chronic inflammation. To prevent ROS-mediated damage, an antioxidant enzyme is released. However, in dermatitis patients, these enzyme-mediated protective responses against ROS are impaired [17,18,19]. Sustained oxidative stress can trigger disease manifestation and carcinogenesis [20,21]. Therefore, we hypothesized that the overexpression of an antioxidant enzyme in MSCs can enhance the anti-inflammatory properties of cells. Superoxide dismutase 3 (SOD3) is one of the members of the superoxide dismutase protein family and is an antioxidant enzyme that catalyzes superoxide radicals. SOD3 dominantly exists in extracellular fluids [22] and protects tissues from oxidative damage [23,24]. SOD3 has been reported to control adaptive immune responses and suppress excessive inflammation in chronic inflammatory skin diseases [25]. We previously found that MSCs expressed undetectable level of SOD3 [26]. These observations led us to investigate whether SOD3 introduction into human umbilical cord blood-derived MSCs (hUCB-MSCs) can augment therapeutic efficacy against dermatitis.

Extracellular vesicles (EVs) are small membrane-bound vesicles that contain proteins, mRNAs, and noncoding RNAs as cargos being transferred to other cells [27,28]. EVs derived from MSCs have been shown to be beneficial for inflammatory diseases [29,30,31,32,33,34]. Recent studies have reported that EVs exhibit similar functional properties to parental cells from which they are produced and have reduced side effects such as immunogenicity or embolisms [35,36]. However, EVs from genetically modified MSCs are less studied, and a large body of data on their characters and functions should be accumulated for the application of EV-based therapeutics.

Therefore, in this study, we investigated the safety and efficacy of hUCB-MSCs introduced with SOD3, a powerful antioxidant enzyme, as well as EVs from these cells. We found that both SOD3-MSCs and their EVs can reduce the symptoms of the murine dermatitis model by attenuating inflammation and fibrosis. These data present strong evidence that SOD3-MSCs and EVs have promising potential for clinical application in dermatitis therapy.

## 2. Materials and Methods

### 2.1. Construction of SOD3-Expressing Lentiviral Vector

Nucleotide sequences encoding the full-length SOD3 (residues 1–240) were first amplified by PCR, using pcDNA-full SOD3 (His6) template, following forward and reverse primers: SOD3-F (BamHI); 5′-CGGGATCCATGCTGGCGCTACTGTGTTC-3′, SOD3-R (XhoI); 5′-GCCTCGAGTTAGG CGGCCTTGCACTCGC-3′. The amplified DNA fragment was inserted into the pLentiM1.9 lentiviral vector, which has murine cytomegalovirus immediate/early promoter as internal promoter (Genetic accession number CR541853). The constructed pLentiM1.9-SOD3 viral vector was confirmed by sequencing analysis (Genotech, Daejeon, Korea).

### 2.2. Lentivirus Production and Transduction

Human embryonic kidney (HEK) 293T cells were cultured in Dulbecco’s Modified Eagle’s medium (DMEM; Gibco, Grand Island, NY, USA) medium with 10% FBS (Gibco) and maintained in 5% CO_2_ incubator at 37 °C. The lentivirus particles were produced by co-transfection of HEK 293T cells with three plasmids: VSV-G, gag-pol, and pLentiM1.9-SOD3, using Lipofectamine Plus (Invitrogen, Carlsbad, CA, USA). At 48 h post-transfection, culture supernatants containing virus particles were collected and clarified with a 0.45 μm membrane filter (Nalgene, Rochester, NY, USA) and stored at −70 °C, in a deep-freezer, immediately. Titers were determined by p24 ELISA (Perkin-Elmer Life Science, Waltham, MA, USA) or Western blot analysis, using a monoclonal anti-p24 antibody (NIH, Bethesda, MD, USA). In our routine preparation, the titers were approximately 1.0 × 10^7^ TU/mL.

### 2.3. hUCB-MSC Culture

This study was approved by the Institutional Review Board of Pusan National University (IRB. No. H-1612-005-004). Human-umbilical-cord-blood-derived mesenchymal stem cells were provided from Kangstem Biotech (Seoul, Korea). Then, hUCB-MSCs were cultured in KSB-3 complete medium (Kangstem Biotech, Seoul, Korea) supplemented with 10% Fetal bovine serum (FBS; Gibco, Grand Island, NY, USA) and 1% Penicillin/Streptomycin (Gibco), at 37 °C, with 5% CO_2_.

### 2.4. Western Blotting

The hUCB-MSC and EV proteins were resuspended by PRO-PREP Protein Extraction Solution (iNtRON Biotechnology, Seoul, Korea). Protein content was qualified by using the BSA protein assay kit (Bio-Rad Laboratories, Hercules, CA, USA). Equal amounts of proteins were loaded and separated on a 12% SDS-PAGE gel and transferred to nitrocellulose membranes (Pall, Port Washington, WI, USA). To prevent non-specific binding, membranes were blocked with 3% BSA at RT for 1 h. Primary antibodies used included anti-SOD3 rabbit polyclonal antibody (Abcam, Cambridge, MA, USA), anti-CD9 rabbit polyclonal antibody (System Biosciences, Palo Alto, CA, USA), anti-CD63 rabbit polyclonal antibody (System Biosciences), anti-CD81 mouse monoclonal antibody (Santacruz, Dallas, TA, USA), anti-HSP70 mouse monoclonal antibody (Abcam), anti-β-tubulin mouse monoclonal antibody (Invitrogen), and anti-GAPDH mouse monoclonal antibody (Invitrogen). Secondary antibodies used were goat anti-rabbit IgG (Invitrogen) and goat anti-mouse IgG (Invitrogen). Proteins were detected by using enhanced chemiluminescence reagent (TransLab, Daejeon, Korea), and bands were imaged by using an ImageQuant LAS 4000 (GE Healthcare, Chicago, IL, USA).

### 2.5. SOD3 Transduction and Detection

The hUCB-MSCs seeded in a 100 mm culture dish at 5 × 10^5^ cells/well were cultured in KSB-3 medium supplemented with 10% heat-inactivated FBS, at 37 °C, for 24 h. The next day, SOD3 lentiviral vector was treated in hUCB-MSCs for 24 h then the medium was replaced with fresh KSB-3 culture medium. 48 h later, culture medium was harvested, and SOD3 level was analyzed by ELISA (R&D system, Abingdon, UK), according to the manufacturer’s instructions, using an ELISA reader, Synergy HTX (BioTek, Winooski, VT, USA). All samples were analyzed in duplicate.

### 2.6. Cell Viability and Proliferation Assay

For cumulative population doubling level (CPDL) analysis, MSCs and SOD3-MSCs plated at 6 × 10^4^ cells/well in 6 well plate were subcultured every 3 days. For each subculture, cell numbers were measured when harvested. The potential of cell proliferation was assessed as CPDL = ln (Nf/Ni) ln2, where Ni = numbers of initially plated cells and Nf = numbers of finally harvested cells. CPDL value at each passage was cumulated to the values of previous passages. HaCaT and HDF were respectively co-cultured with hUCB-MSCs or treated EVs in 24 well plates at 3 × 10^4^ cells/well. Cell viability was measured at 24 h. A solution from Cell Counting Kit-8 (CCK-8; Dojindo Molecular Technologies, Rockville, MD, USA) was supplemented into cell culture media, followed by measurement of the absorbance at 450 nm, using a microplate reader, Bio-Tek Synergy HTX.

### 2.7. hUCB-MSC Differentiation

#### 2.7.1. Osteogenic Differentiation

hUCB-MSCs were plated at the density of 5 × 10^3^ cells/cm^2^ in a 12-well plate. Medium was changed when the confluency reached up to 70%. The osteogenic medium was composed of complete DMEM supplemented with 10% heat-inactivated FBS, 100 nM dexamethasone, 50 μM ascorbic acid, and 10 mM beta-glycerophosphate (Sigma, Saint Louis, MO, USA). The medium was changed twice a week. After 2 weeks of differentiation induction, cells were washed twice with PBS, fixed in 4% paraformaldehyde (PFA; Biosesang, Seongnam, Korea) for 15 min, at room temperature, and stained with 2% Alizarin Red S (Kanto, Tokyo, Japan) for 1 h. After rinsing cells twice with PBS, the extracellular matrix calcification was visualized under the microscope.

#### 2.7.2. Adipogenic Differentiation

The hUCB-MSCs were plated at the density of 1 × 10^4^ cells/cm^2^, in a 12-well plate. The medium was changed when the confluency reached up to 90%. The adipogenic medium was composed of complete DMEM supplemented with 10% heat-inactivated FBS, 1 μM dexamethasone, 3-isobutyl-1-methylxanthine, and 0.2 mM indomethacin (all from the Sigma). The medium was changed twice a week. After 3 weeks of induction, cells were washed twice with PBS, fixed in 4% PFA for 15 min, at room temperature, and stained with 2% Oil Red O (Sigma) for 1 h. After rinsing cells twice with PBS, the intracellular lipid droplets were visualized under the microscope.

### 2.8. Flow Cytometric Analysis

Cells were stimulated with 50 ng/mL phorbol 12-myristate 13-acetate (PMA; Sigma) and 1 μM ionomycin (Sigma) with transport inhibitor GolgiStop (BD Biosciences, San Jose, CA, USA) in a cell incubator with 5% CO_2_, at 37 °C, for 4 h. Cells were transferred to tubes and washed with PBS. Cells were stained with anti-human CD4 antibodies conjugated with FITC (BD Biosciences, San Jose, CA, USA) for 30 min, at 4 °C. After fixation and permeabilization, cells were incubated with anti-human IFNγ antibodies conjugated with PerCP-Cy5.5, anti-human IL-4 antibodies conjugated with APC, and anti-human FoxP3 antibodies conjugated with PerCP-Cy5.5 (BD Biosciences, San Jose, CA, USA) for 30 min, at 4 °C. Isotype controls were used to correct nonspecific binding. Cells were analyzed with BD FACS Canto II flow cytometer (Becton, Dickinson and Company, Franklin Lakes, NJ, USA), and data were analyzed with FlowJo software (Becton, Dickinson and Company).

### 2.9. Cytokine Array

Culture supernatant from MSCs and SOD3-MSCs was collected, centrifuged to remove cell debris, and stored at −80 °C. The expression level of immunomodulation- and migration-related cytokines or growth factors in the culture supernatant was detected, using C-Series Human Cytokine Array (Raybiotech, Norcross, GA, USA), according to the manufacturer’s instructions.

### 2.10. Mixed Lymphocyte Reaction and Mitogen-Stimulated Proliferation Assay

Human peripheral blood mononuclear cells (PBMCs) were incubated with carboxyfluorescein diacetate (CFDA-SE; Invitrogen), at 37 °C, for 15 min, and washed with complete RPMI 1640 medium (Gibco). The hUCB-MSCs and human PBMCs were co-cultured at a 1:10 ratio, with 5 × 10^4^ hUCB-MSCs for 5 × 10^5^ human PBMCs per mL. Positive control cells were treated with 25 μl/mL anti-CD3/28. After 5 days, cells were analyzed with BD FACS Canto II flow cytometer (Becton, Dickinson and Company), and data were analyzed with FlowJo software (Becton, Dickinson and Company).

### 2.11. Isolation and Culture of Human CD4^+^ Helper T Cells

This study was approved by the Institutional Review Board of Pusan National University (IRB. No. H-1802-006-063). Cord blood was provided from Dong-A University Hospital Cord Blood Bank (Busan, Korea). Mononuclear cells were isolated from human cord blood, using HetaSep (Stem Cell Technologies, Vancouver, Canada). Total CD4^+^ T cells were purified by negative selection, with magnetic beads (Human Naive CD4^+^ T Cell Isolation Kit II; Miltenyi Biotec, Bergisch Gladbach, Germany). Purified CD4^+^ T cells were plated in the density of 1 × 10^6^ cells/well in 12-well culture plates. Cells were cultured in ImmunoCult-XF T Cell Expansion Medium (Stem Cell Technologies) supplemented with ImmunoCult Human CD3/CD28/CD2 T Cell Activator (Stem Cell Technologies) and recombinant human interleukin-2 (IL-2; Peprotech, Rocky hill, NJ, USA), for 7 days, at 37 °C, with 5% CO_2_.

### 2.12. CD4^+^ T-Cell Polarization

Purified CD4^+^ T cells were co-cultured with hUCB-MSCs or exosome from hUCB-MSCs. Cells were plated at a 1:10 ratio (CD4^+^ T cells: hUCB-MSCs or CD4^+^ T cells: Exosome from hUCB-MSCs) in 24-well culture plates. Th1 cells were polarized in ImmunoCult -XF T Cell Expansion Medium supplemented with ImmunoCult Human CD3/CD28/CD2 T Cell Activator, 10 ng/mL recombinant human interleukin-12 (IL-12; Peprotech), and 5 μg/mL anti-IL-4 neutralizing antibodies (BD Bioscience) for 5 days. Th2 cells were polarized in ImmunoCult-XF T Cell Expansion Medium supplemented with ImmunoCult Human CD3/CD28/CD2 T Cell Activator, 20 ng/mL recombinant human interleukin-4 (IL-4; Peprotech), and 5 μg/mL anti-IFNγ neutralizing antibodies (BioxCell, Lebanon, NH, USA) for 5 days. Th1 and Th2 were treated with 20 ng/mL IL-2 on day 3 of polarization. Treg cells were polarized in RPMI 1640 supplemented with 10% FBS, 2 mM glutamax (Gibco), 50 μM β-mercaptoethanol, 2 ng/mL transforming growth factor beta 1 (TGF-β1; Peprotech), and 10 ng/mL IL-2 for 5 days.

### 2.13. Macrophage Activation Assay

THP-1 cells were obtained from Korean Cell Line Bank (Seoul, Korea). THP-1 cells seeded in 6-well plates, at 4 × 10^5^ cells/well, were cultured in RPMI 1640 supplemented with 10% heat-inactivated FBS and stimulated with PMA, at 37 °C, for 24 h. After media replacement with normal media, THP-1 derived macrophages were stabilized for 48 h and polarized in M1 type, using LPS (10 μg/mL, InvivoGen) and IFN-γ (20 ng/mL, Sigma). On day 3, hUCB-MSCs-derived EVs or SOD3-MSCs derived EVs were treated in THP-1-derived macrophages. After 24 h, culture media was harvested, and the TNF-α level was analyzed by ELISA (R&D system) according to the manufacturer’s instructions, using an ELISA reader, Bio-Tek Synergy HTX. THP-1 cells were seeded with hUCB-MSCs or SOD3 transduced MSCs, at 6 wells, in RPMI 1640 media with 10% FBS (MSC:THP-1, 1:10). Cultured media was harvested after 5 days of co-culture, and TNF-α level was measured by using ELISA. All assays were performed in duplicate. The concentration was calculated by using a linear-regression equation obtained from the standard absorbance values.

### 2.14. LAD-2 Cell Culture

Human mast cell line, LAD-2 cells, were cultured in StemPro-34 SFM medium (Gibco) supplemented with 1% Penicillin/Streptomycin and 100 ng/mL recombinant human stem cell factor (SCF; Peprotech). LAD-2 cells were plated at a concentration of 5 × 10^5^ cells/mL maintained, at 37 °C, with 5% CO_2_. Cell culture medium was hemidepleted every week with fresh medium.

### 2.15. LAD-2 Cell Degranulation and β-Hexosaminidase Analysis

To induce LAD-2 cell degranulation, cells plated in 24-well plates, at 3 × 10^5^ cells/well, were cultured in StemPro-34 SFM medium supplemented with 1% penicillin/streptomycin and 100 ng/mL SCF and sensitized with 100 ng/mL IgE (Millipore, Burlington, MA, USA), at 37 °C, with 5% CO_2_. After 24 h, cells were treated with 3 μg/mL anti-IgE (Millipore) for challenge. The hUCB-MSCs or EVs were added with anti-IgE challenge. After the incubation for 1 h, the plate was placed on ice to stop degranulation, and supernatants were harvested. The supernatant was incubated with an equal volume of substrate solution (7.5 mM p-nitrophenyl-*N*-acetyl-β-*D*-glucosaminide dissolved in 0.05 M citrate buffer, pH = 4.5). The mixture was incubated on a shaking incubator, for 2 h, at 37 °C. The reaction was terminated by the addition of 0.2 M NaOH/0.2 M glycine. The absorbance was measured at 410 nm, using a plate reader. The β–hexosaminidase release was calculated as the percentage of total β-hexosaminidase content in naïve mast cells, measured after cell lysis with 0.1% Triton X-100.

### 2.16. HaCaT Cell and Human Dermal Fibroblast (HDF) Culture

HaCaT cells (ATCC, Manassas, VA, USA) and HDF cells (Korean Cell Line Bank) were cultured in DMEM medium (Gibco) supplemented with 10% heat-inactivated FBS and 1% Penicillin/Streptomycin, at 37 °C, with 5% CO_2_.

### 2.17. Immunofluorescence

HDFs were seeded at the density of 2 × 10^4^ cells/well, in 6-well culture plates, and allowed to adhere for 6 h. HDFs were treated with 10 ng/mL TGF-β1 in DMEM serum free media and then co-cultured with MSCs or SOD3-MSCs for 48 h. For cell staining, HDFs were fixed in 4% PFA for 20 min, at room temperature. HDFs were washed three times with PBS, followed by incubation with 0.1% Triton X-100, for 10 min, at room temperature. HDFs were washed three times with PBS and blocking with 5% Goat serum (Vector, Burlingame, CA) for 1 h, at room temperature. HDFs were incubated with anti-SOD3 primary antibodies, overnight, at 4 °C. HDFs were washed three times with PBS, and they were incubated with Alexa Fluor 488-conjugated secondary antibody (Santa Cruz) for 1 h, at room temperature. HDFs were washed three times with PBS, the nuclei were stained with Hoechst 33342 (Abcam) for 10 min, at room temperature, and then mounted for analysis, under a fluorescence microscope (Carl Zeiss Meditec, Jena, Germany).

### 2.18. Chloro-2, 4-dinitrochlorobenzene (DNCB)-Induced Mice Model

Male BALB/c mice at 8–10 weeks were obtained from Jackson Laboratory (Bar Harbor, ME, USA). The day before the experiment, the back of the mouse was shaved. DNCB model was induced in mice by multiple treatment of DNCB (Sigma) in vehicle (acetone:olive oil = 3:1 mixture). Briefly, 1% DNCB solution was treated on the dorsal skin and induced an immune response (sensitization) for 3 days, followed by treatment with 0.2% DNCB solution on the dorsal skin, for 10 days, to induce contact dermatitis (challenge, Appendix A). The average of the individual scores (0 = none; 1 = mild; 2 = moderate; 3 = severe) on skin dryness, excoriation, erythema, and edema was calculated to determine the clinical severity.

### 2.19. Migration Assay

#### 2.19.1. In Vitro

The hUCB-MSCs were seeded in the upper chamber of an 8 μm pore transwell (Greiner Bio-One, Kremsmünster, Austria) at 1 × 10^5^ cells/mL. On the other side, IFNγ/TNFα and LAD-2 CM were loaded into the lower chamber. After 24 h, migrated cells were fixed with 4% PFA for 10 min and stained with crystal violet for 20 min. Cells in random separate microscope fields were counted.

#### 2.19.2. In Vivo

In order to evaluate the distribution of hUCB-MSCs, dorsal skin tissues from designated area (A and B, Appendix A) at 14 days were harvested, and gDNA was isolated. The quantification of hUCB-MSCs was performed by Real-Time PCR for human ALU transcripts. The primer sequences were as follows: 5′-GTC AGG AGA TCG AGA CCA TCC C-3′ (ALU Forward) and 5′-TCC TGC CTC AGC CTC CCA AG-3′ (ALU Reverse). DNA of mouse skin was mixed with 0%, 0.004%, 0.016%, 0.031%, 0.062%, 0.125%, 0.25%, 0.5%, 1%, 2%, 4%, 8%, and 16% of g DNA from hUCB-MSCs and standard curve was obtained based on *Cp* values determined in mixed samples. The hUCB-MSC concentration in each harvested sample is calculated based on the standard curve.

### 2.20. Histological Analysis

Dorsal skin tissues were collected from the AD-induced mice for histological analysis on day 14. Tissues were fixed in 10% PFA solution and embedded in paraffin. Sections from the mouse dorsal skins were stained by using hematoxylin and eosin (H&E), to determine changes in the thickness of the epidermis or toluidine blue to indicate the changes in mast cell infiltration.

### 2.21. Isolation of EVs from Conditioned Medium of hUCB-MSCs

The hUCB-MSCs were cultured in KSB supplemented with 10% FBS for 24 h at 37 °C with 5% CO_2_. Cells were washed twice with PBS when they reached 80% confluence and then cultured for 72 h in KSB supplemented with 10% exosome-depleted FBS. The supernatants were centrifuged at 3000× *g* for, 15 min, to remove cell debris and centrifuged at 100,000× *g* (Beckman Coulter, Brea, CA, USA), for 70 min, at 4 °C. The supernatants were discarded, and the pellets were washed in PBS, re-suspended, and centrifuged at 100,000× *g* for, 70 min, at 4 °C. The pellet was re-suspended in 100 μL of PBS or protein extraction solution and stored at −80 °C.

### 2.22. EV Uptake Assay

To verify whether the EVs are internalized into HaCaT and HDF cells, cells were cultured overnight at a density of 1 × 10^4^ cells/cm^2^ in 24 well culture plates. EVs from hUCB-MSCs were labeled with PKH67 green fluorescent cell linker kit (Sigma) according to the manufacturer’s instruction. Labeled EVs were treated in HaCaT and HDF for 24 h, and the nuclei were stained with Hoechst 33342. Images were obtained by using a Zeiss LSM 700 confocal microscopy system (Carl Zeiss Meditec).

### 2.23. Statistical Analysis

Results were analyzed using GraphPad Prism software (GraphPad, San Diego, CA, USA). Statistical evaluation was performed by two-tailed Student’s *t*-test or one-way ANOVA, followed by Bonferroni post hoc test for multi-group comparisons. Results are presented as means ± SD.

## 3. Results

### 3.1. Vector Integration in hUCB-MSCs

The hUCB-MSCs were genetically engineered to produce SOD3, using lentiviral transduction. A schematic diagram of the vector is shown in Figure 1A. This lentiviral vector was used to transduce SOD3 into hUCB-MSCs at different multiplicity of infection (MOI). Cell morphology was similar in naïve cells and SOD3-transduced cells (Figure 1B). To evaluate the transduction efficiency, we detected the level of SOD3 expression by immunoblotting. The expression of SOD3 protein in hUCB-MSCs was elevated in an MOI-dependent manner (Figure 1C). We next measured the level of secreted SOD3 in the supernatant of cell culture by ELISA. SOD3 concentration in culture media was increased MOI-dependently and significantly elevated at 5 and 10 MOI (Figure 1D). Transduced cells were determined for the vector copy number (VCN). VCN was increased in an MOI-dependent manner, and the number was similar to MOI value (Figure 1E). We selected 5 MOI for further experiments since SOD3-MSC transduced at 5 MOI showed a significant increase in SOD3 expression or secretion with VCN less than 10. The hUCB-MSCs from two different donors were confirmed for reproducible SOD3 expression and VCN (Figure 1F,G). These results suggest that the transduction of SOD3 efficiently achieves sustained production of SOD3 protein in hUCB-MSCs.

### 3.2. Characterization of SOD3-Transduced MSCs

We investigated whether SOD3 transduction influences characteristics of hUCB-MSCs. The proliferation of naïve or engineered cells was determined. Cell proliferation was slightly increased after SOD3 transduction in both MSC#1 and MSC#2 (Figure 2A). In addition, SOD3 overexpression did not alter the differentiation potential of hUCB-MSCs into osteoblasts or adipocytes (Figure 2B). To detect whether transduction might alter the pattern of surface marker expression, well-known markers to validate MSCs and their immunogenicity were determined by flow cytometry. Non-transduced and transduced MSCs showed a similar pattern of surface antigen expression (Figure 2C; Appendix A). To secure the genomic stability of genetically engineered cells, we performed karyotyping in SOD3-transduced cells and observed no chromosomal abnormalities in both transduced MSCs (Figure 2D). To identify the change in the production of various soluble factors involved in the immunomodulation and migration of hUCB-MSCs after transduction, a cytokine array was conducted. The hUCB-MSCs produced high levels of interleukin (IL)-8, IL-6, thrombospondin (TSP)-1, and tissue inhibitor of metalloproteinase (TIMP)-2 (Appendix A). Among immunomodulatory paracrine factors, the levels of galectins, ALCAM, B7-H1, and Fas ligand were slightly increased in MSCs after SOD3 overexpression (Figure 2E). These results demonstrate that transduced cells maintain the original characteristics of MSCs.

### 3.3. Immunomodulatory Effects of SOD3-MSCs on Human Immune Cells

To determine whether SOD3-MSCs are immunogenic or immunosuppressive, we co-cultured MSCs with human mononuclear cells (hMNCs) in the absence or presence of stimulation for T cells and measured the proliferation of hMNCs. The proliferation of hMNCs was increased when co-cultured with hUCB-MSC; however, its level was significantly lower compared to anti-CD3/28-stimulated hMNCs. Of interest, SOD3-transduced hUCB-MSCs exhibited weak induction of hMNCs proliferation compared with naïve MSCs (Figure 3A). When T-cell proliferation was stimulated with anti-CD3/28, both MSCs or SOD-MSCs significantly inhibited the proliferation of T cells to a similar extent (Figure 3B). We next explored whether SOD3-MSCs can affect the differentiation of CD4^+^ T cells. CD4^+^ T lymphocytes were co-cultured with MSCs or SOD3-MSCs under either condition for Th1 or Th2 differentiation, and the proportion of IFN-γ^+^ and IL-4^+^ cells was measured. MSCs inhibited the differentiation of Th1 cells, and SOD3 transduction increased this inhibitory effect. In particular, inhibition of Th1 differentiation by MSC#1 was significantly augmented by SOD3 overexpression (Figure 3C). This difference observed in the potency of immunoregulation among MSCs might be caused by individual variabilities in MSCs from variable donors. In Th2 differentiation, MSCs, and SOD3-MSCs exhibited a similar suppressive pattern (Figure 3D). Given that MSCs can induce the generation of regulatory T cells (Tregs), we evaluated whether SOD3-MSCs can exhibit Treg-generating potency. CD4^+^ T cells were co-cultured with MSCs or SOD3-MSCs, without inducing Treg differentiation, and Foxp3^+^ cells were detected. SOD3-MSCs induced the generation of Treg cells more efficiently than MSCs (Figure 3E). To further determine the regulatory abilities of SOD3-MSCs on immune cells other than T cells, macrophage-like cells and mast cell line (LAD-2) were co-cultured with MSCs. The addition of both MSCs or SOD-MSCs similarly abrogated the production of TNF-α from activated macrophage-like cells (Figure 3F). MSCs were added at sensitization and challenge step of mast cell degranulation to investigate their regulatory abilities on mast cells (Appendix A). Interestingly, SOD3 overexpression significantly increased the inhibitory effect of MSCs on LAD-2 cell degranulation (Figure 3G; Appendix A). Taken together, these data suggest that SOD3-transduction in hUCB-MSCs can augment anti-inflammatory properties, particularly through improved suppression of Th cell differentiation and mast cell degranulation.

### 3.4. Anti-Fibrotic Feature of SOD3-MSCs

Itch-induced scratching in atopic dermatitis leads to severe scar formation and regulation of fibrosis, as well as regeneration in wound healing, is important. Therefore, we next investigated whether SOD3-MSCs could facilitate the proliferation of skin cells or suppress the differentiation of fibroblasts into myofibroblasts. Co-culture experiments using transwell revealed that MSCs and SOD3-MSCs did not affect the proliferation of HDF and HaCaT cells (Figure 4A,B). To determine anti-fibrotic effects, TGF-β1 was treated in HDF, to induce myofibroblast differentiation, and MSCs were co-cultured. Immunofluorescence staining with α-smooth muscle actin (SMA) revealed that MSCs and SOD-MSCs can significantly suppress myofibroblast generation (Figure 4C). Our findings indicate that SOD3-MSCs possess anti-fibrotic features which might be beneficial for the later phase of AD therapy.

### 3.5. Therapeutic Efficacy of SOD3-MSCs in Murine Dermatitis Model

To assess the therapeutic potency of SOD3-MSCs in vivo, we established a DNCB-induced mouse model for AD-like dermatitis, and MSCs were subcutaneously infused on day 7 (Figure 5A). On day 14, clinical severity was scored. MSC administration significantly decreased the symptoms of AD, and SOD3-MSC further ameliorated symptoms to a greater extent (Figure 5B). Upon histopathological examination, DNCB treatment-induced epidermal hyperplasia and the infiltration of inflammatory cells. The MSC-treated group exhibited thinner epidermis and lower lymphocyte infiltration compared to vehicle (PBS)-treated group. SOD3 transduction augmented this therapeutic efficacy. In particular, SOD3-MSCs significantly attenuated lymphocyte infiltration (Figure 5C,E,F). Toluidine blue staining showed that mast cell infiltration in the lesion was increased in the DNCB-treated group, which was prevented by MSC or SOD3-MSC administration. Especially, SOD3-MSC#1 injection significantly reduced the infiltration of mast cells, presumably due to donor-dependent differences of MSCs’ abilities on mast cell regulation (Figure 5D,G). Our findings suggest that SOD3-MSCs can exhibit potent therapeutic efficacy against atopic dermatitis in vivo.

### 3.6. Migration of SOD3-MSCs to the Inflamed Site

Given that the migratory abilities of MSCs toward the inflamed area are pivotal for their therapeutic efficacy, we evaluated whether SOD3 transduction might alter the migration capacity of MSCs. Using transwell, in vitro migration MSCs toward IFN-γ/TNFα or conditioned media (CM) from LAD-2 cells were analyzed. MSCs migrated toward the inflammatory cues and SOD3 overexpression did not influence the migration capability (Figure 6A). More importantly, we next evaluated the distribution of SOD3-MSCs in vivo by measuring the expression of human-specific ALU gene in skin tissues harvested from injection sites (A) and inflamed sites (B) (Appendix A). After 2 or 24 h of MSC administration, most of the cells were detected in the injection site of skin and gradually diminished at days 3 and 7. Interestingly, at day 3, more cells were detected in the inflamed site compared to day 1, indicating that MSCs migrate to the lesion where inflammation is excessive (Figure 6B; Appendix A). However, there was no significant difference in the distribution pattern between MSC- and SOD3-MSC-injected groups. These results imply that SOD3-MSCs possess similar migratory properties and distribution patterns as MSCs have.

### 3.7. Recapitulation of SOD3-MSC Functions by EVs

Since EVs from various MSCs have been reported to recapitulate some of the beneficial functions of their parental cells, we next investigated whether SOD3-MSCs derived EVs express SOD3 and can exert similar diverse regulatory functions. We generated SOD3-transduced MSCs and isolated EVs from the cells (Figure 7A). EVs were characterized by detecting CD9, 63, 81, and HSP70 (Figure 7B). Interestingly, SOD3 protein was expressed in EVs from SOD3-MSCs (SMSC-EVs) (Figure 7B). Immunoregulatory and anti-fibrotic abilities observed in SOD3-MSCs were evaluated in EVs. While SOD3-MSCs did not augment the inhibitory effect of MSCs against T-cell proliferation, interestingly, SMSC-EVs more efficiently downregulated the T-cell proliferation, as compared with MSC-EVs (Figure 7C). SMSC-EVs parallelly exert suppressive potency in CD4^+^ T cell differentiation, as their parental cells did (Figure 7D,E). Significant regulation of SOD3-MSC#1 on Th1 differentiation was consistently observed in EVs from the cells, indicating that donor-dependent individual variabilities can also be recapitulated in EVs. Moreover, Treg generation was similarly induced by the treatment of both MSC-EVs and SMSC-EVs (Figure 7F). However, while MSCs almost completely abrogated the activation of macrophages regardless of SOD3 overexpression (Figure 3F), EVs did not recapitulate this phenotype (Figure 7G). On mast cell degranulation, the inhibitory effect of MSC-EVs on degranulation was slightly decreased in SMSC-EVs (Figure 7H). We next examined whether SMSC-EVs can exhibit regeneration-accelerating or anti-fibrotic abilities. No significant changes were observed in the proliferation of HDF or HaCaT when treated with MSC-EVs or SMSC-EVs (Figure 7I,J). Myofibroblast differentiation was suppressed by MSC- or SMSC-EV treatment (Figure 7K). Finally, we determined whether SMSC-EVs can deliver SOD3 protein into other cells. PKH67-labeled EVs were treated in fibroblast, and EV incorporation into fibroblast was detected. Fluorescence-labeled EVs were detected in all cells. SOD3 protein expression was detectable only in SMSC-EV-treated cells, indicating that EVs can deliver SOD3 into other cells (Figure 7L). To further investigate whether EVs show a preference to a certain type of cells in incorporation efficiency, labeled EVs were treated to co-cultured cells (keratinocyte + T cell or mast cell + T cell). Interestingly, EVs incorporation efficiency was high in keratinocytes and moderate in T cells or mast cells (Appendix A). Taken together, these observations suggest that SOD3-MSC-derived EVs recapitulate the T-cell-regulatory and anti-fibrotic abilities of MSCs and that EVs can deliver the targeted protein into other cells.

### 3.8. In Vivo Efficacy of SOD3-MSC-Derived EVs in Dermatitis Model

To finally confirm the therapeutic efficacy of SMSC-EVs, EVs were treated in DNCB-induced dermatitis mice and symptoms were analyzed according to the same timeline utilized in MSC transplantation experiment (Figure 8A). Clinical severity was significantly ameliorated by subcutaneous administration of SMSC#1-EVs (Figure 8B). Moreover, SMSC-EVs significantly suppressed the thickening of the epidermis and the infiltration of lymphocytes (Figure 8C,E,F). Although mast cell infiltration was not significantly downregulated by MSC- or SMSC-EV treatment, SMSC#1-EVs showed higher inhibitory potency, as compared to other groups (Figure 8D,G). These findings indicate that EVs from SOD3-transduced MSCs can efficiently alleviate the symptoms of AD and that variability in therapeutic potency of MSCs from different donors can be recapitulated in EVs.

## 4. Discussion

In the present study, we demonstrated that transduction of SOD3 in hUCB-MSCs improved the immunoregulatory and therapeutic functions of cells and their EVs against in vitro immune cell activation and in vivo dermatitis manifestation, respectively. We found that hUCB-MSCs expressed almost undetectable level of SOD3 protein and viral transduction of the gene remarkably overexpressed its protein in an MOI-dependent manner. Overexpression of SOD3 in hUCB-MSCs did not alter the characteristics of MSCs as stem cells. MSCs exert anti-inflammatory effects mainly through the production of various soluble factors such as TGF-β1, LIF, HGF, galectins, and chemokines for immune cells [37,38,39,40,41,42,43]. Moreover, their migration toward inflamed areas is also important to efficiently modulate local inflammatory responses. In this study, we observed the expression of pivotal factors related to immunomodulation and migration of MSCs and found that the production pattern of well-known factors in MSCs was not dramatically changed by the overexpression of SOD3 on the protein level.

The pathogenesis of atopic dermatitis is initiated by type 2 helper T cell (Th2)-mediated acute inflammatory responses, followed by Th1-mediated chronic phase [44,45,46]. Degranulation of MCs contributes to the progression toward the chronic phase, by recruiting lymphocytes and granulocytes [47]. More recently, macrophages have been reported to be involved in the pathogenesis of atopic dermatitis [48]. In our study, co-culture of these AD-inducing immune cells with naïve MSCs resulted in the downregulation of cell proliferation, maturation, and activation. MSCs significantly suppressed the proliferation of T cells, as well as the differentiation of Th1 and Th2 subsets, while they induced the generation of Tregs. Moreover, MSCs decreased the TNF-α production from macrophage-like cells and the degranulation of MSCs. SOD3 overexpression augmented these suppressive effects of MSCs on the activation of various immune cell subtypes, except for macrophages. Especially, MSC#1 exhibited more significant synergistic effects when transduced with SOD3 upon the regulation of Th1 differentiation. This in vitro regulatory function of SOD3-MSCs correlated with the superior therapeutic efficacy of modified MSCs in vivo. These findings are consistent with our previous studies demonstrating superior regulatory abilities of SOD3-MSCs on T-cell subsets and MCs, as well as improved therapeutic efficacy against psoriasis and atopic dermatitis [13,26]. Although previous studies mostly utilized immune cells from mice, in the present study, we used human-mononuclear-cell-derived T cells to confirm the regulatory mechanisms in human cell-to-cell interaction.

Itch is one of the hallmarks of AD, and scratching contributes to the acceleration of disease symptoms and scar formation [49,50]. In the later stage of wound repair, dermal fibroblasts remodel the wound matrix and uncontrolled differentiation of fibroblasts into myofibroblasts leads to permanent scar-tissue formation, which also impairs skin regeneration [51]. Therefore, regulation of skin regeneration is also important following the resolution of inflammation. Since several studies have demonstrated anti-fibrotic functions of MSCs [52,53], in this study, we analyzed this anti-fibrotic function of naïve or SOD3-MSCs. Consistent with previous reports, MSCs significantly downregulated the myofibroblast differentiation, regardless of SOD3 transduction.

Although immunoregulatory properties of MSCs against various immune disorders have been shown, MSC-based therapy confronts several limitations to overcome: quality control as a living cell, high cost, and low-reproducibility [54,55]. Here we showed that further genetic modification can improve functional potency to overcome low-reproducibility, but the quality-control issue still remains. Therefore, cell-free therapeutics using derivatives from MSCs have been spotlighted as a novel strategy. Moreover, EVs can be a perfect candidate because EVs reflect the characteristics of parent cells. Recently, a number of studies have shown that MSC-derived EVs contain various proteins, mRNAs, and microRNAs produced in MSC, which can exert similar anti-inflammatory abilities of MSCs. EVs also have advantages in that handling and storage can be much easier, and safety issues, including embolism and immunogenicity, observed after cell transplantation, can be minimized [56,57]. However, since the quantity of proteins in EVs is very small, most studies have focused on mRNA or miRNA to elucidate the mechanism of anti-inflammatory effects of MSC-EVs [58,59,60,61,62]. In the present study, we demonstrated that overexpressed protein in MSCs can be delivered by EVs and protein-carrying EVs can introduce the protein into other cells. Moreover, these EVs recapitulated some of the immunoregulatory phenotypes of parental cells. Interestingly, EVs showed the preference of incorporation according to the type of cells. Further studies are required to determine the location of SOD3 protein in EVs and EV incorporation preference into different cell types, as well as to elucidate more detailed mechanism in synergistic effects of SOD3 with already known miRNAs.

In our results, SOD3-MSC#1 and their EVs generally exhibited superior immunosuppressive or therapeutic potency compared to SOD3-MSC#2 and EVs (Figure 3C, Figure 5G, Figure 7C,D, and Figure 8B,E). These donor-dependent individual differences in MSCs have been suggested as one of the crucial limitations in the clinical application of MSCs as cell therapeutics, and several studies have tried to uncover underlying mechanisms for these variabilities [63]. A number of studies have shown that MSCs from different donors exhibit significant differences in growth, differentiation, and paracrine functions, with distinguished expressions of relevant genes [64,65,66]. Although these previous studies tried to explore the group of markers elucidating the mechanism of differences in cellular functions, individual variabilities might not be fully revealed by previously analyzed factors. Therefore, a large body of further investigations is required to establish the optimized and selected pool of screening factors for each cellular function based on the accumulated data from a genome-wide association study or secretome analysis.

EVs from genetically modified MSCs, as well as modified MSCs themselves, have a number of advantages for clinical application. SOD3-MSCs might lead to successful outcomes in clinical trials since SOD3-introduced MSCs possess potent immunomodulatory abilities required for AD alleviation, as compared to naïve MSCs, and SOD3 enzyme is expected to prolong the survival of MSCs after transplantation, by attenuating damage from ROS in the lesion. However, in our experimental settings, we could not observe the improved distribution of SOD3-MSCs in inflamed areas, as compared to non-transduced MSCs. One might envision that xenograft transplantation of MSCs in immunocompetent mice led to strong immune responses, and SOD3-mediated protection of harsh environments, including high ROS, was attenuated. EVs exhibit more superior advantages in that they can overcome the physical or immunological hindrance of cell transplantation, represented by embolism or short clearance of MSCs, respectively. However, a major limitation in EV-based therapy is the low quantity of EVs from cells, using current isolation technologies. Therefore, future studies should focus on developing technologies for improved production and isolation of EVs, which carry highly enriched proteins and miRNAs that are beneficial for the treatment of diseases.

Taken together, our findings suggest that SOD3 overexpression in MSCs results in the improved immunomodulatory abilities of T-cell immunity, as well as the amelioration of in vivo symptoms observed in the murine dermatitis model. Moreover, EVs from modified cells deliver SOD3, which also can efficiently regulate the excessive inflammation in vitro and in vivo. Our findings might provide a novel strategy for the development of EV-based therapeutics.

## Figures and Tables

**Figure 1 antioxidants-09-01165-f001:**
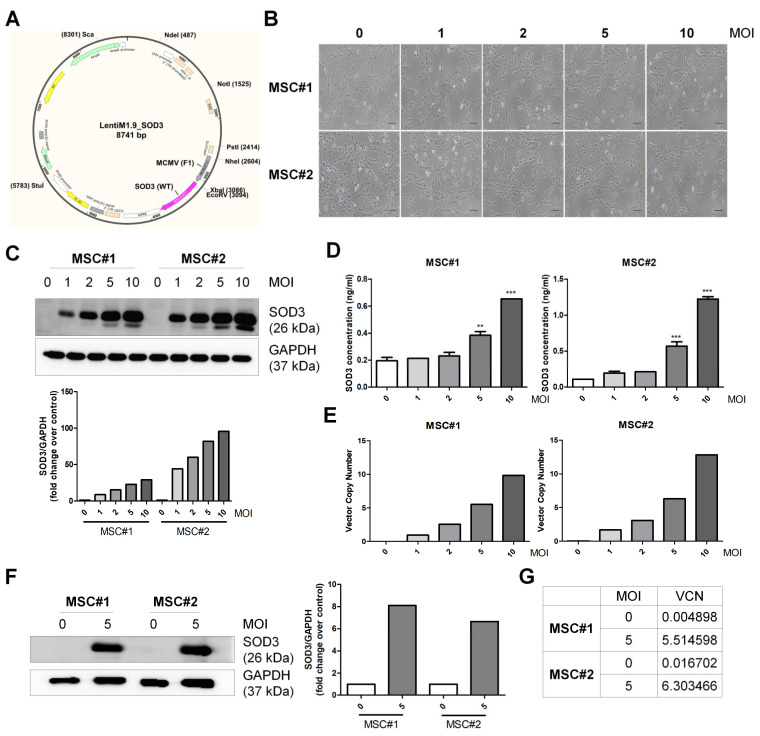
Validation of SOD3 transduction in human umbilical cord blood-derived mesenchymal stem cells (hUCB-MSCs). (**A**) Schematic representation of the lentiviral vector (containing MCMV promoter) for the expression of SOD3 (gene). (**B**) Morphology SOD3-transduced MSC#1, MSC#2 cells taken at 48 h post-infection, at various multiplicity of infection (MOI). Scale bar = 100 μm. (**C**) Detection and quantization of SOD3 in protein extracts of non-transduced and transduced MSCs at 0, 1, 2, 5, and 10 MOI. (**D**) Measurement of secreted SOD3 in culture media of MSCs at 48 h post-infection, by ELISA. (**E**) Average vector copy number (VCN) in transduced MSCs at 72 h post-infection. Genomic DNA from MSCs was extracted and used for average VCN measurement, using quantitative PCR assay. (**F**,**G**) Confirmation of SOD3 overexpression and VCN in transduced MSCs at 5 MOI. Results are shown as mean ± SD. ** and *** represent *p* < 0.01 and *p* < 0.001, respectively.

**Figure 2 antioxidants-09-01165-f002:**
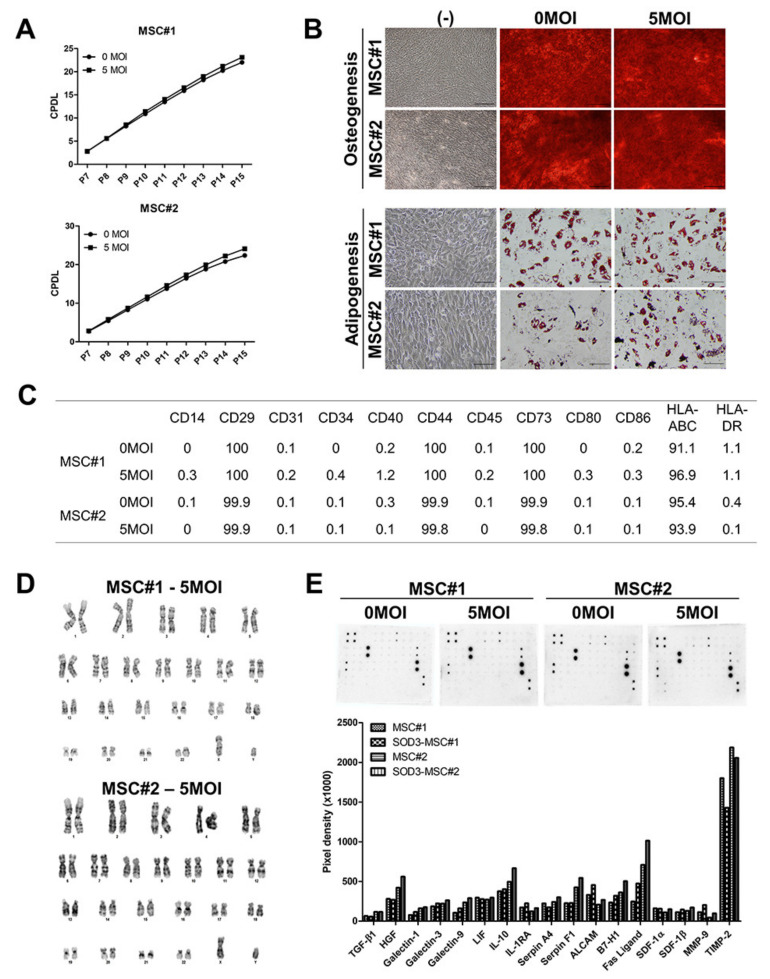
MSC characterization after SOD3 transduction. (**A**) Cumulative population doubling level (CPDL) of MSCs through continuous passaging from P7 to P15. (**B**) MSCs were induced to differentiate into osteoblasts and adipocytes for two and three weeks and were stained by Alizarin Red S and Oil Red O, respectively. Scale bar = 100 μm. (**C**) Analysis of surface marker expression in MSCs by flow cytometry. (**D**) Karyotyping of SOD3-transduced MSCs. (**E**) Cytokine array of MSC-derived conditioned media. Quantification of the pixel density for each identified cytokine in MSCs. Data are normalized after background subtraction.

**Figure 3 antioxidants-09-01165-f003:**
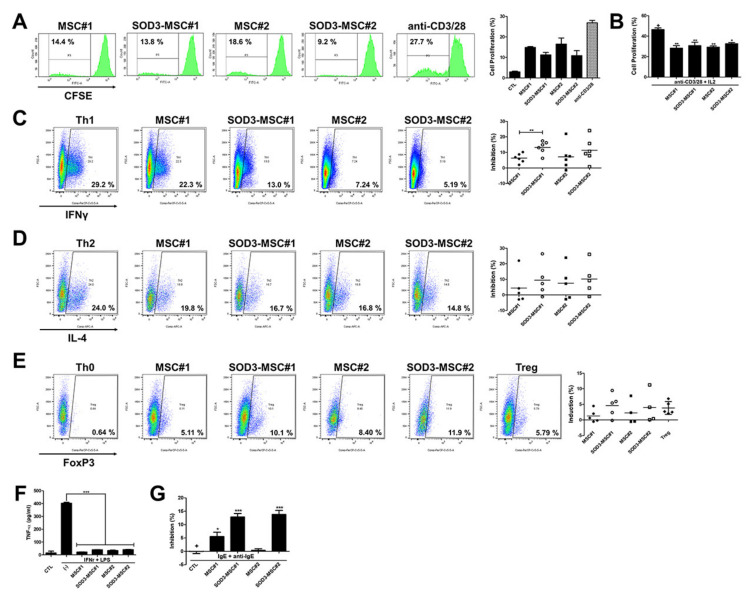
The immunomodulation of SOD3-MSCs. (**A**) The peripheral blood mononuclear cell (PBMC) proliferation without any stimulants was detected by CFSE assay in the presence of MSCs (MSC:PBMC = 1:10), using transwell co-culture. Proliferation in the positive control group was induced with anti-CD/28. (**B**) Proliferation of anti-CD3/CD28 activated CFSE-labelled CD4^+^ T cells co-cultured with MSCs or SOD3-MSCs. (**C**–**E**) Representative dot plots and accumulated values showing the percentages of Th1 (**C**), Th2 (**D**), and Treg (**E**) cells in CD4^+^ T cells co-cultured with MSCs or SOD3-MSCs. (**F**) THP-1 derived macrophages were activated and co-cultured with MSCs or SOD3-MSCs. TNF-α level in culture media was detected by ELISA. (**G**) β-hexosaminidase content was measured in the supernatants from activated LAD-2 cells. Results are shown as mean ± SD. *, **, and *** represent *p* < 0.05, *p* < 0.01, and *p* < 0.001, respectively. *p*-value significance was calculated by comparing other groups against the (+) group (marked as +). ●: MSC#1, ○: SOD3-MSC#1, ■: MSC#2 and □: SOD3-MSC#2.

**Figure 4 antioxidants-09-01165-f004:**
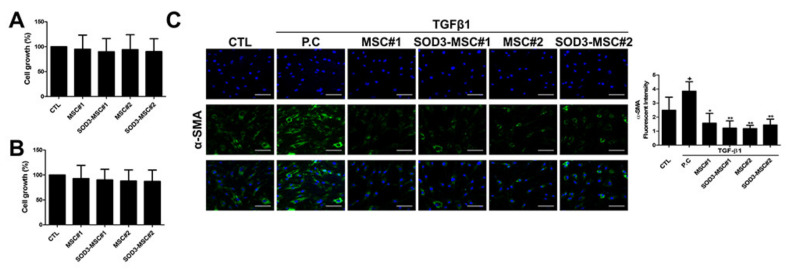
The regenerative and wound-healing potential of MSCs. (**A**,**B**) Effects of SOD3-MSCs on human dermal fibroblast (HDF) and HaCaT proliferation in transwell co-culture system were examined by CCK-8 assay. (**C**) Representative images of α-SMA expression. HDF was cultured with 10 ng/mL TGF-β for 48 h in the absence or presence of MSCs and α-SMA (green) was detected by immunofluorescence. α-SMA level was quantified by Image J. Scale bar = 100 μm. Results are shown as mean ± SD. * and ** represent *p* < 0.05 and *p* < 0.01, respectively. *p*-value significance was calculated by comparing other groups against the (+) group (marked as +).

**Figure 5 antioxidants-09-01165-f005:**
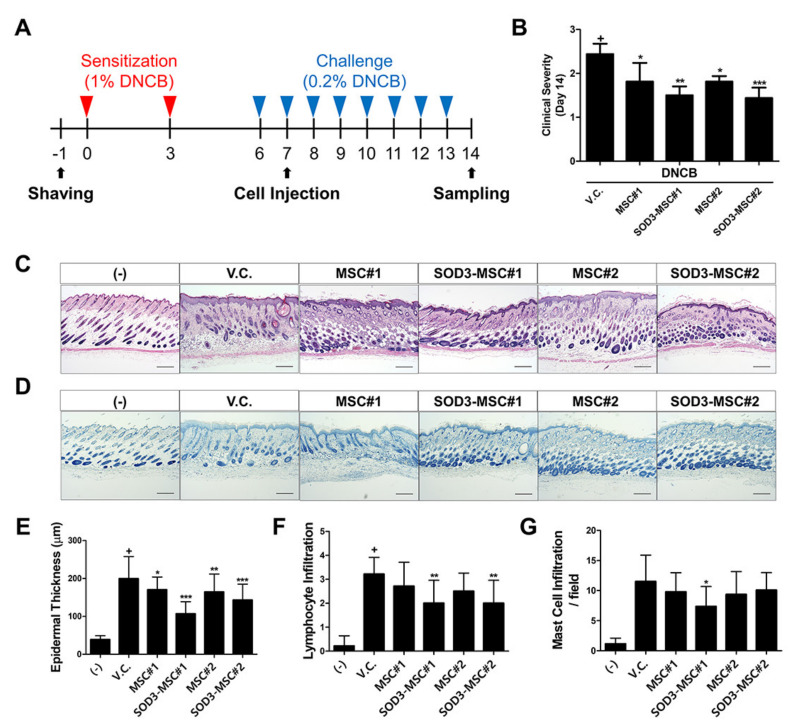
In vivo anti-inflammatory effects of MSCs. (**A**) 2, 4-dinitro chlorobenzene (DNCB) was used to induce atopic dermatitis (AD) in BALB/c mice. MSCs or SOD3-MSCs were injected subcutaneously, near the lesion site on day 7. (**B**) On day 14, the clinical severity of mice was scored. V.C.; vehicle control. (**C**) Skin tissues of mice were harvested after sacrifice and were subjected to hematoxylin and eosin (H&E) staining. Scale bar = 100 μm. (**D**) Toluidine blue staining was performed to determine the number of mast cells. Scale bar = 100 μm. (**E**) Epidermal thickness was measured. (**F**,**G**) The number of infiltrated lymphocytes (**F**) and mast cells (**G**) was counted. Results are shown as mean ± SD. *, **, and *** represent *p* < 0.05, *p* < 0.01, and *p* < 0.001, respectively. *p*-value significance was calculated by comparing other groups against the (+) group (marked as +).

**Figure 6 antioxidants-09-01165-f006:**
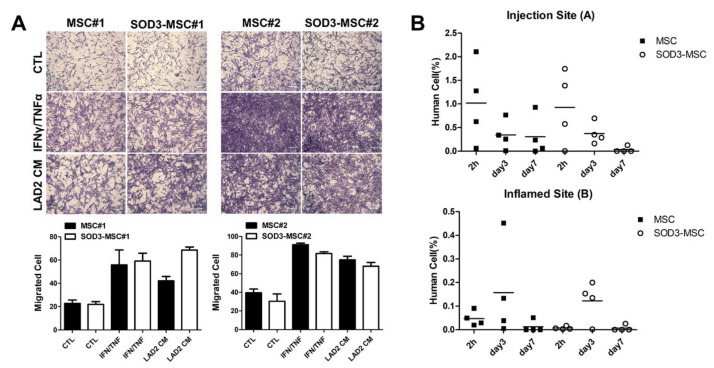
In vitro and in vivo migration of MSCs toward the inflammatory site. (**A**) Migration of MSCs and SOD3-MSCs in response to inflammatory factors was determined in vitro, using transwell. Representative photographs of migrated MSCs or SOD3-MSCs in the bottom of the upper chamber. Scale bar = 100 μm. (**B**) The amount of MSCs in the injection site and inflamed site at 2, 72, and 168 h after injection, using qRT-PCR for human ALU sequence. Results are shown as mean ± SD.

**Figure 7 antioxidants-09-01165-f007:**
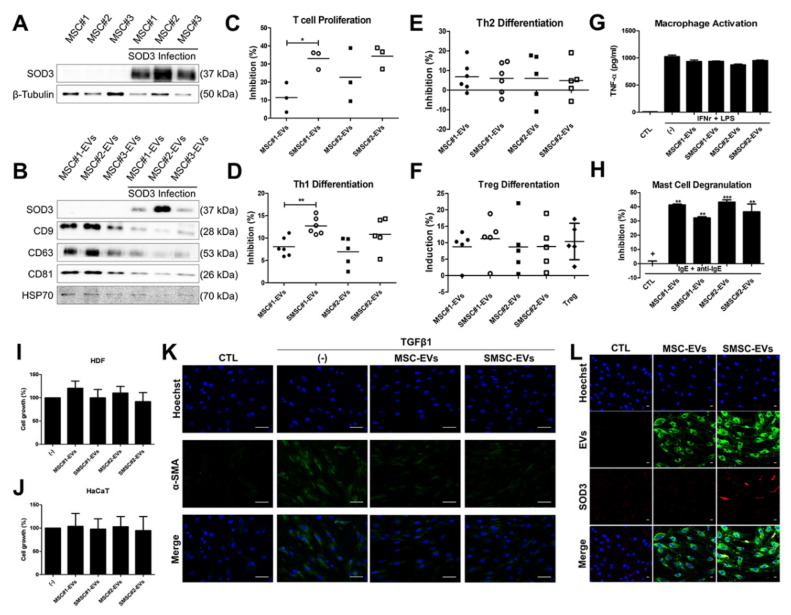
Immunoregulatory and anti-fibrotic features of extracellular vesicles (EVs) from MSCs. (**A**) Detection of SOD3 protein in SOD3-transduced MSC#1, MSC#2, and MSC#3 (at 48 h, 5 MOI) by immunoblotting. (**B**) Expression of SOD3 protein and EV markers (CD9, 63, 81, and HSP70) in SOD3-MSC-derived EVs. (**C**) T-cell-proliferation assay. (**D**–**F**) Representative dot plots and accumulated values showing the changes in percentages of Th1 (**D**), Th2 (**E**), and Treg (**F**) cells in CD4^+^ T cells treated with EVs. (**G**) THP-1 derived macrophages were activated and TNF-α production in the absence or presence of EVs was detected by ELISA. (**H**) β-hexosaminidase levels released into the culture media were measured by ELISA. (**I,J**) CCK-8 assay for HDF and HaCaT-cell proliferation after EV treatment for 24 h. (**K**) Myofibroblast differentiation and its inhibition by EV treatment were determined by α-SMA staining. Scale bar = 100 μm. (**L**) HDFs were incubated with EVs for 24 h and EV uptake (green) and SOD3 protein delivery (red) were detected by confocal microscopy. Scale bar = 20 μm. Results are shown as mean ± SD. *, ** and *** represent *p* < 0.05, *p* < 0.01 and *p* < 0.001, respectively. *p*-value significance was calculated by comparing other groups against the (+) group (marked as +). ●: MSC#1, ○: SOD3-MSC#1, ■: MSC#2 and □: SOD3-MSC#2.

**Figure 8 antioxidants-09-01165-f008:**
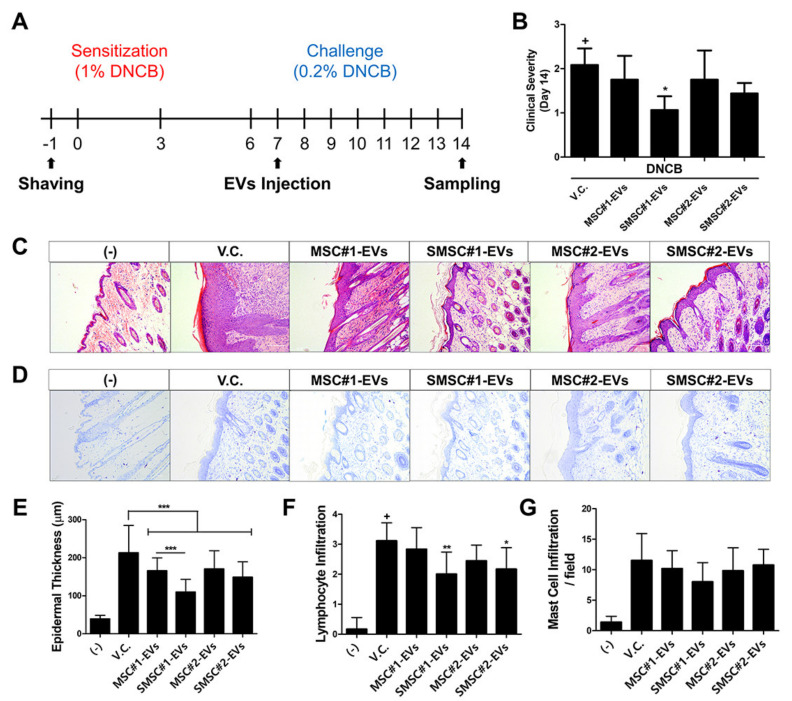
Therapeutic efficacy of EVs derived from SOD3-MSCs in AD mice. (**A**) MSC-derived EVs were subcutaneously administrated into DNCB-induced AD mice on day 7. (**B**) On day 14, clinical severity was scored (**C**) H&E staining and (**D**) toluidine blue staining of skin tissue sections. (**E**) Epidermal thickness, (**F**) lymphocyte infiltration, and (**G**) mast cell infiltration were measured. Scale bar = 100 μm. Results are shown as mean ± SD. *, ** and *** represent *p* < 0.05, *p* < 0.01 and *p* < 0.001, respectively. *p*-value significance was calculated by comparing other groups against the (+) group (marked as +).

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
