# Peer review of "Extracellular Vesicles from SOD3-Transduced Stem Cells Exhibit Improved Immunomodulatory Abilities in the Murine Dermatitis Model"

_antioxidants, 2020, doi:10.3390/antiox9111165_

Round 1

Reviewer 1 Report

The article "Extracellular Vesicles from SOD3-transduced Stem 2 Cells Exhibit Improved Immunomodulatory Abilities 3 in the Murine Dermatitis Model" by Ji Won Yang is a very well compiled and executed study and is of high merit. I see no immediate and major issues with the paper besides some very minor typographical errors (i.e. CO2 is superscripted in some appearances and not in other instances). In all summary, this is a solid manuscript and can be published.

Author Response

Authors’ response: We would like to thank this reviewer for positive and insightful comments on our study. We checked and corrected typographical errors throughout the manuscript including the one that this reviewer indicated.

Reviewer 2 Report

Yang et al. describe the generation of SOD3-transduced stem cells and their impact on immunomodulatory activities in co-culture model and mouse model.  The research methods are well described and a number of aspects of SOD3-transduced MSC were explored including extracellular vesicles derived.  I believe that the authors can improve their manuscript as follows:

1. There are significant differences in the immunomodulatory activities of MSC#1 and MSC#2 such that only one of them are statistically significant for some assays. The authors need to discuss these variability at length in the result and discussion sections.  It would also help cite more relevant papers regarding the high variability observed. 

2. There are a number of apparent typos in the manuscript. E.g., to generation -> to generate (line 39), CD4 T^+ cell -> CD4^+ T cell (line 193), In vivo efficay -> in vivo efficacy (line 501).  The authors need to carefully go through and edit the manuscript. It would also help to get English editing services.    

Author Response

Reviewer #2: Yang et al. describe the generation of SOD3-transduced stem cells and their impact on immunomodulatory activities in co-culture model and mouse model.  The research methods are well described and a number of aspects of SOD3-transduced MSC were explored including extracellular vesicles derived.  I believe that the authors can improve their manuscript as follows:

  1. There are significant differences in the immunomodulatory activities of MSC#1 and MSC#2 such that only one of them are statistically significant for some assays. The authors need to discuss these variability at length in the result and discussion sections. It would also help cite more relevant papers regarding the high variability observed. 

Authors’ response: We appreciate the reviewer’s comment. We think that the reviewer suggested an important point. As suggested, we added comment for each result showing the donor-dependent variability and wrote a paragraph in discussion to deal with this issue. And 4 more references were added for citation. Following is newly added paragraph in discussion with references.

--------------------------------------------------------------------------------------------

(p.16, line 565-575)

In our results, SOD3-MSC#1 and their EVs generally exhibited superior immunosuppressive or therapeutic potency compared to SOD3-MSC#2 and EVs (Fig 3C, 5G, 7C and D, 8B and E). This donor-dependent individual differences in MSCs have been suggested as one of crucial limitations in clinical application of MSCs as cell therapeutics and several studies tried to uncover underlying mechanisms for this variabilities [64]. A number of studies have shown that MSCs from different donors exhibit significant differences in growth, differentiation and paracrine functions, with distinguished expressions of relevant genes [65-67]. Although these previous studies tried to explore the group of markers elucidating the mechanism of differences in cellular functions, individual variabilities might not be fully revealed by previously analyzed factors. Therefore, a large body of further investigations are required to establish the optimized and selected pool of screening factors for each cellular function based on the accumulated data from genome wide association study or secretome analysis.

[64] G. Siegel, T. Kluba, U. Hermanutz-Klein, K. Bieback, H. Northoff, R. Schafer, Phenotype, donor age and gender affect function of human bone marrow-derived mesenchymal stromal cells, BMC Med 11 (2013) 146.

[65] I. Kang, B.C. Lee, S.W. Choi, J.Y. Lee, J.J. Kim, B.E. Kim, D.H. Kim, S.E. Lee, N. Shin, Y. Seo, H.S. Kim, D.I. Kim, K.S. Kang, Donor-dependent variation of human umbilical cord blood mesenchymal stem cells in response to hypoxic preconditioning and amelioration of limb ischemia, Exp Mol Med 50(4) (2018) 35.

[66] D.G. Phinney, G. Kopen, W. Righter, S. Webster, N. Tremain, D.J. Prockop, Donor variation in the growth properties and osteogenic potential of human marrow stromal cells, J Cell Biochem 75(3) (1999) 424-36.

[67] R. Siddappa, R. Licht, C. van Blitterswijk, J. de Boer, Donor variation and loss of multipotency during in vitro expansion of human mesenchymal stem cells for bone tissue engineering, J Orthop Res 25(8) (2007) 1029-41.

  1. There are a number of apparent typos in the manuscript. E.g., to generation -> to generate (line 39), CD4 T^+ cell -> CD4^+ T cell (line 193), In vivo efficay -> in vivo efficacy (line 501).  The authors need to carefully go through and edit the manuscript. It would also help to get English editing services.    

Authors’ response: We appreciate the reviewer’s elaborate review on our manuscript. As suggested, we checked and corrected typographical errors throughout the manuscript including those that the reviewer indicated. All corrected typos are highlighted in red font.

Reviewer 3 Report

I’ve read with attention the paper of Ji Won Yang et al. that is potentially of interest. The background and aim of the study have been clearly defined. The methodology applied is overall correct, the results are reliable and adequately discussed. However, the discussion could be further enriched with some comments on the possible clinical application of the observed results, on the possible research perspectives in this field and on the possible limitation of the applied methodologies.

Author Response

Reviewer #3: I’ve read with attention the paper of Ji Won Yang et al. that is potentially of interest. The background and aim of the study have been clearly defined. The methodology applied is overall correct, the results are reliable and adequately discussed. However, the discussion could be further enriched with some comments on the possible clinical application of the observed results, on the possible research perspectives in this field and on the possible limitation of the applied methodologies.

Authors’ response: We would like to thank this reviewer for positive and insightful comments on our study. And we think that the reviewer suggested an important point regarding clinical application and limitation of methodologies. As suggested, we added a paragraph in discussion to deal with this issue. Following is newly added part in discussion.

----------------------------------------------------------------------------------------------------------------

(p.17-18, line 576-589)

EVs from genetically modified MSCs, as well as modified MSCs themselves, have a number of advantages for clinical application. SOD3-MSCs might lead to successful outcomes in clinical trials since SOD3-introduced MSCs possess potent immunomodulatory abilities required for AD alleviation compared to naïve MSCs and SOD3 enzyme is expected to prolong the survival of MSCs after transplantation by attenuating damage from ROS in the lesion. However, in our experimental settings, we could not observe the improved distribution of SOD3-MSCs in inflamed area compared to non-transduced MSCs. One might envision that xenograft transplantation of MSCs in immunocompetent mice led to strong immune responses and SOD3-mediated protection of harsh environment including high ROS was attenuated. EVs exhibit even superior advantages in that they can overcome physical or immunological hindrance of cell transplantation, represented by embolism or short clearance of MSCs, respectively. However, a major limitation in EV-based therapy is the low quantity of EVs from cells using current isolation technologies. Therefore, future studies should focus on developing technologies for improved production and isolation of EVs, which carry highly enriched proteins and miRNAs, beneficial for the treatment of diseases.